# "One feels anger to know there is no one to help us!". Perceptions of mothers of children with Zika virus-associated microcephaly in Caribbean Colombia: A qualitative study

Elena Marbán-Castro[1]*, Cristina Enguita-Fernàndez[1], Kelly Carolina Romero-Acosta[2], Germán J. Arrieta[2,3,4], Anna Marín-Cos[1], Salim Mattar[3,4], Clara Menéndez[1,5,6], Maria Maixenchs[1,5], Azucena Bardají[1,5,6]

1 ISGlobal, Hospital Clínic—Universitat de Barcelona, Barcelona, Spain, 2 Corporación Universitaria del Caribe (CECAR), Group of Public Health, Sincelejo, Sucre, Colombia, 3 Clínica Salud Social, Sincelejo, Sucre, Colombia, 4 Universidad de Córdoba—Instituto de investigaciones biológicas del Trópico, Montería, Córdoba, Colombia, 5 Consorcio de Investigación Biomédica en Red de Epidemiología y Salud Pública (CIBERESP), Madrid, Spain, 6 Centro de Investigação em Saúde de Manhiça (CISM), Maputo, Mozambique

* elena.marban@isglobal.org

**Data Availability Statement:** This is a qualitative study based on a sample of women who had a

## Abstract

### Background

The epidemic of Zika virus (ZIKV) was associated with a sudden and unprecedented increase in infants born with microcephaly. Colombia was the second most affected country by the epidemic in the Americas. Primary caregivers of children with ZIKV-associated microcephaly, their mothers mainly, were at higher risk of suffering anxiety and depression. Often, these women were stigmatized and abandoned by their partners, relatives, and communities.

### Methodology/Principal findings

This study aimed to understand the perceptions about ZIKV infection among mothers of children born with microcephaly during the ZIKV epidemic in Caribbean Colombia, and the barriers and facilitators affecting child health follow-up. An exploratory qualitative study, based on Phenomenology and Grounded Theory, was conducted in Caribbean Colombia. Data were collected through In-Depth Interviews (IDI) from women who delivered a baby with microcephaly during the ZIKV epidemic at Clínica Salud Social, Sincelejo, Sucre District (N = 11). The themes that emerged during the interviews included experiences from their lives before pregnancy; knowledge about ZIKV; experiences and perceptions when diagnosed; considering a possible termination of pregnancy, and children's clinical follow-up. In some cases, women reported having been told they were having a baby with microcephaly but decided not to terminate the pregnancy; while in other cases, women found out about their newborn's microcephaly condition only at birth. The main barriers encountered by participants during children's follow-up included the lack of psychosocial and economic support,

child with microcephaly associated with Zika virus infection during pregnancy, in a very specific area in Caribbean Colombia. Making the data fully available would breach their privacy. The informed consent that all women signed promised full anonymity. Additionally, data cannot be made publicly available to a repository as this goes against the Research Ethics Committee [Comité de Bioética Institucional de la Universidad de Córdoba, Montería (Colombia)]. Relevant excerpts of the data are displayed within the manuscript. Following data requests, transcripts will be reviewed for any potentially identifying information and will only be made available to researchers who sign a data-sharing agreement. Data inquiries may be directed to the Biostatistics and Data Management Unit at ISGlobal <ubioesdm@isglobal.org>.

**Funding:** This study was funded by the Government of Spain under grant agreement number PI16/0123, ISCIII-AES Proyectos de Investigación en Salud, 2016; granted to A.B. Additionally, E. M-C. was supported by a predoctoral fellowship from "la Caixa" Foundation (ID 100010434); fellowship code LCF/BQ/ES17/ 11600006. E.M-C. received a mobility grant by Universitat de Barcelona for her stay in Colombia. A.B. was supported by the Ministry of Science, Innovation, and Universities, Government of Spain through a Ramon y Cajal Grant (RYC-2013-14512). ISGlobal is a member of the CERCA Programme, Generalitat de Catalunya. We acknowledge support from the Spanish Ministry of Science and Innovation through the "Centro de Excelencia Severo Ochoa 2019-2023" Program (CEX2018-000806-S), and support from the Generalitat de Catalunya through the CERCA Program. The funders had no role in study design, data collection and analysis, decision to publish, or preparation of the manuscript.

**Competing interests:** The authors have declared that no competing interests exist.

the stigmatization and abandonment by some partners and relatives, and the frustration of seeing the impaired development of their children.

## Conclusions

This study contributed to identifying the social, medical, psychological, and economic needs of families with children affected by the ZIKV epidemic. Commitment and action by local and national governments, and international bodies, is required to ensure sustained and quality health services by affected children and their families.

## Author summary

As of January 2018, nearly 4000 cases of children with congenital birth defects associated with Zika virus infection, included microcephaly, had been reported in the Americas. In 2019, we interviewed eleven mothers of children with microcephaly in Colombia, a country heavily affected by the Zika epidemic. Most women were young, lived in peri-urban areas were of low socioeconomic status, according to official government classification in Colombia. They reported knowing that the virus was transmitted by mosquitoes, and how to prevent mosquito bites, yet they were not aware of Zika infection being transmitted by sexual contact. Most women had been tested for Zika during pregnancy but did not receive their laboratory results. Also, they complained that they had not received enough information from healthcare providers. When fetal anomalies were detected prenatally, only a few of them were offered the possibility to terminate the pregnancy. Women reported a lack of both economic and psychosocial support to deal with the special health care needs their children required. The results of this study highlight the struggles and challenges of families of children severely affected by the Zika virus and provide valuable information for health policy making as part of the next epidemic preparedness and response.

## Introduction

In 2015–17, the epidemic of Zika virus (ZIKV) in the Americas caused a sudden increase in children born with microcephaly and other neurological manifestations associated with severe brain lesions such as brain disruption, microcalcification, eye damage, hypertonia, and arthrogryposis [1,2]. This spectrum of signs and symptoms was described under the term Congenital Zika Syndrome (CZS) [1,2]. CZS includes a diverse pattern of structural brain anomalies and functional disabilities in affected children, secondary to central nervous system damage [3]. Microcephaly is its most common sign, observed in 80% of CZS cases [3]. ZIKV infection, especially in the first trimester of pregnancy, was associated with severe outcomes in fetuses and children [4]. The consequences of CZS not only involve affected children, but their environment, including their community, relatives, and their primary caregivers, usually their mothers, who are more prone to suffer anxiety and depression due to difficulties in providing the specialized care their children require [5–10]. Mothers of children affected by microcephaly experience significant emotional and psychological distress; as most of them are stigmatized, abandoned by their partners, relatives, and communities [5]. Additionally, it has been reported that their mental health worsens over the years [6,11]. Families were also affected by changes produced in the daily routine, which in turn, impacts the couple's relationship, the

possibility to work, as sometimes one parent needs to leave the job for child care, leading to economic difficulties [12]. Most affected families are from low socioeconomic status according to official government classification in Colombia, where ZIKV cases are more prevalent, then, experiencing more restrictions to provide specialized care [5]. Repercussions of having a baby with CZS have been reported to exacerbate social inequalities and poverty [9]. Even when specialized assistance for children was received, women did not have psychosocial support to cope with this stressful situation [13].

According to WHO reports, Colombia was the second country with the highest prevalence of ZIKV cases, and CZS babies in the American region, after Brazil [14]. In Colombia, the epidemic led to three to four times as many infants born with microcephaly in 2015–2017 compared to estimates expected in the absence of the virus [15]. Among all ZIKV cases in Colombia, 66% of them were women [16]; and its burden fell on the poorest populations [17]. However, a study performed in Colombia found a lower prevalence of ZIKV cases in poorer areas, but authors hypothesize that it could be due to the limited access of healthcare centres in poor areas to report cases, decreased inter-municipal connectivity in those places, and lower access to education, reducing medical attendance and case detection [18]. The real costs to cover the specialized care that microcephalic children require are unknown, and might be life-long. Though it is very difficult to estimate the annual cost of having a ZIKV-associated microcephalic child, a study in Brazil estimated that the healthcare treatment was 5,937 USD per child, per year [19]. ZIKV infection was most prevalent in young women that lived in the outskirts of the cities, and in low-socioeconomic statuses [20]. In Colombia, the average monthly income is 441$, and the poverty rate (population living on less than 5.5$/day) for 2018 was 27% [21]. According to official government classification in Colombia, districts are rated from 1 to 6 for its affluence. Populations living in districts of Status 1 and 2 are considered poor, and are eligible for an insurance financed by the government, entitled 'Subsidized health Regimen' (régimen subsidiado) [22,23]. However, being affiliated to this regimen has been associated with poorer health outcomes, due to social determinants of these populations, and barriers in access to healthcare services [24].

In 2016, the national Government implemented the ZIKV Fever Response Plan to strengthen public health surveillance systems, health education, vector control strategies and encouraged research and close monitoring of pregnant women and children's health [16]. On the 7[th] of January 2016, the Ministry of Health published a notice in which they recommended: "*All couples living in the country, don't get pregnant during the epidemic phase, until July 2016 [. . .] and pregnant women who do not live in an area below 2.000 msnm, not to travel to those areas for the high risk of acquiring the infection, until July 2016*" [25]. Regarding sexual and reproductive health rights (SRHR), ten years before, in 2006, advocacy efforts contributed to the partial decriminalization of abortion in Colombia in three instances: when pregnancy constitutes a threat to the woman's health or life, if the fetus presents anomalies incompatible with life, or if pregnancy is caused by rape or incest [26,27]. Still, lack of access, makes abortion the fifth leading cause of maternal mortality in the country [26,27]. A qualitative study performed with key informants from the Colombian national and local governments, healthcare providers, community members, and affected women, concluded that a multidimensional approach that considers healthcare services, gender issues, and people's environment is crucial for the success of ZIKV campaigns; and that the effects on women's rights are related to inequalities in SRHR (increased risk of sexually transmitted infections by the poorest and most vulnerable women) [16].

Structural gender inequities in Central and South America are well documented, and women often are not in control of their reproductive decisions [17]. Populations most affected by ZIKV often had limited access to reproductive health services, such as antenatal care,

contraception counselling, and services -including emergency contraception-, safe abortion, and post-abortion care [28,29]. Many countries affected by ZIKV still have restrictive abortion laws, making safe abortion completely illegal or very difficult to access [29]. Furthermore, the epidemic did not change the voluntary abortion landscape in any relevant way. In fact, in Brazil, a law was introduced to increase jail sentences for women seeking abortion "*due to microcephaly or other foetal anomaly*" [30]. In some countries, such as El Salvador and Colombia, authorities recommended sexual abstinence as a preventive measure against ZIKV infection [27,31]. As policies left women solely with the responsibility to avoid pregnancy, in countries where abortion is criminalized, unsafe abortions could have led to rises in maternal morbidity and mortality [5].

ZIKV was the most recent infection disproportionately affecting women in different aspects of health and wellbeing, including their emotional state and mental health. Yet, the social, economic, cultural, and personal consequences of the epidemic remain unknown; and its magnitude represents a challenge. Furthermore, the effects and ravages that Zika caused in Colombian families, are unknown. The Ebola, Cholera, Zika, and COVID-19 outbreaks raised the need to understand the social pathways of disease transmission and barriers affecting populations at risk [32,33]. The role of Anthropology in emerging outbreaks has raised global awareness for the integration of sociocultural approaches in response to international health crises [32,33]. The present study aimed to explore the views, perceptions, and attitudes, towards ZIKV, and challenges, including barriers and facilitators to medical follow-up of children born with microcephaly in the ZIKV epidemic in Caribbean Colombia, faced by their primary caregivers: their mothers.

## Methods

### Ethics statement

Ethical approval for the study was granted by the Committee of the Universidad de Córdoba, Montería [Reg. No. FMVZ-001-2016] and by the study Clínica Salud Social in Sincelejo Ethics Committee [Reg. No. F-GI-IV-001]. The study was conducted following the Good Clinical Practice Guidelines and under the provisions of the Declaration of Helsinki and local rules and regulations. Participants gave written consent for the interviews to take place and be audio-recorded. All names in the transcripts were deleted to guarantee subject anonymity. All names appearing in the manuscript have been created.

### Study design

This exploratory qualitative study used a phenomenological approach to understanding first-hand experiences of affected populations [34]. Grounded theory was used as a methodological and analytical approach to inductively create theoretical generalizations that emerge from the data [35].

### Study site and population

The study was performed between April and July 2019. Participants were identified from a study conducted in rural and peri-urban areas in Córdoba and Sucre departments, Caribbean Colombia. That study included a clinical follow-up of microcephalic children born at the study hospital "Clínica Salud Social" in Sincelejo (Sucre) during the first semester of 2016 [11,36]. Mothers of followed-up children were contacted by phone calls to participate in the current study. Inclusion criteria for study participants was defined as: having a newborn diagnosed with microcephaly during the ZIKV epidemic in Colombia, and being willing to be

interviewed and audio-recorded as part of study procedures. Interviews took place in the departments of Córdoba, Sucre, Bolívar, and Cartagena, according to women's place of residence and preference.

## Data collection

Data collection took place from April to July 2019. Data were collected through In-Depth Interviews (IDI) with primary caregivers (mothers) of children born with microcephaly in 2016. When interviews occurred, children had two or three years of age, and in most of the cases, they were present during the interview. The duration of IDIs was around 45–60 minutes. Interviews were conducted at participants' place of preference, including health facilities (Clínica Salud Social in Sincelejo), participant´s places of residence, or public spaces. All interviews were digitally recorded and notes were taken.

## Data analysis

Interviews were transcribed and data were coded using Dedoose software (SocioCultural Research Consultants, LLC, Manhattan Beach, CA, USA). Consensus on codes and emerging themes were reached in meetings within the investigators´ team. Research began with no preexisting hypothesis, allowing theories to inductively emerge from the data, following a systematic and circular data collection and analysis, according to Grounded Theory [35]. The theory generation was based on comparative analyses among data collected from different participants, and pre-existing conceptualizations were not used [34,37]. Overlapping themes and subthemes emerging from the participants' narratives are summarized in S1 Table.

## Results

### Participants' profiles

A total of eleven women whose children were born with microcephaly born during the first semester of 2016 were enrolled in the study. The average age of participants was of 26 years. All female caregivers were the biological mothers of the microcephalic children, except for one who was the grandmother of the child whose biological parents refused to take care of the baby. All women gave birth by a C-section. Table 1 describes the sociodemographic characteristics of study participants.

### Knowledge of ZIKV and sources of information

Knowledge of ZIKV was very diverse and related to the educational background of the women interviewed. Some women had a very basic knowledge of the virus, with gaps in knowledge in the understanding of Zika being a virus, transmission and consequences of the disease, expressed by short sentences such as "*Zika is a disease that. . . is what is happening to my child, is like a bacteria that affects children, is the only thing that I know*" (Jenny, 20 years old), or "*with the fever you have, the rash in the body, with that it is transmitted*" (Nadia, 21 years old). Others had very good knowledge and understanding of the virus and the disease, even including technical terms in their narratives such as "*brain anomalies, microcephaly, eye defects*" as a consequence of infection during pregnancy, or mentioned, "*hemorrhagic Dengue infection*" i.e. "*It's a mild disease [Zika], generally, temporary, as happened to me. It causes pain, and once one is pregnant, it affects very much the brain and well, the normal system of the baby*" (Paola, 23 years old). Knowledge was generally acquired by the TV, the Internet, public health campaigns, knowledge from people inside the community, and their own experience. While some women changed their behaviour with mass media (i.e. stop watching TV) to protect

**Table 1. Sociodemographic characteristics of women participating in the study, ZIKV-associated symptoms during pregnancy, and description of microcephaly diagnosis.**

| Name* | Age | Education | Occupation | Relatives living with her | Religion | SES **** | ZIKV symptoms (pregnancy month) | How was maternal ZIKV/ microcephalic children being diagnosed? |
|---|---|---|---|---|---|---|---|---|
| **María** | 32 | Secondary school | Child care | Partner, daughter, and microcephalic child | Christian | Low | Rash, muscular pain, and headache (3) | All ultrasound (US) exams were normal. The only concern was that the foetus was underweight. In the last US (month 8), foetal head measurement did not correlate with other anthropometric parameters. |
| **Nadia** | 21 | Primary school | Child care | Partner, mother in law, father in law, grandmother in law, and microcephalic child | Christian | Low | Fever, headache and vomiting (early in pregnancy), and does not remember if she presented with any rash. | All US assessments were normal, until the last one, when head circumference was detected to be smaller than expected for gestational age. |
| **Adelaida** | 26 | Practical training | Child care | Partner, daughter, and microcephalic child | Christian | Low | Rash and fever (3) | She was tested for ZIKV, never received her results. All US were normal, until the last one when "microcephaly" was diagnosed. |
| **Carla** | 30 | Unknown | Unknown | Partner, and microcephalic child | Christian | Low | Rash (2) | All US exams were normal. At month 7, head circumference was small for gestational age. At month 8, microcephaly was confirmed. |
| **Guadalupe** | 18 | University (ongoing) | Student | Mother, two siblings, and microcephalic child | Christian | Low | Rash (before realizing she was pregnant) | First US exam was normal. Patient didn´t attend to second US appointment due to monetary constraints. During third US, measures did not correlate with gestational age, and a C-section was performed. |
| **Judith** | 29 | University | Teacher (literates' adults) | Partner, and microcephalic child | Christian | Medium | Rash, fever, and bone pain (not defined) | All US exams were normal. Only concern clinicians had was that the baby had intrauterine growth restriction. |
| **Georgina** | 43 | Secondary | Child care | Partner, child, daughter-in-law, two grandchildren, and the grandchildren with microcephaly (She was the grandmother but primary caregiver)*** | Christian | Low | Biological mother presented fever (2 or 3) | All US exams were normal. At month 8, microcephaly was confirmed. |
| **Paola** | 23 | University | Child care | Father, mother, brother, niece, and microcephalic child | Christian | Low | Rash, fever, and body pain (1) | In an US exam (month 6) microcephaly and Dandy–Walker syndrome were diagnosed in the baby. |
| **Jenny** | 20 | Primary school | Assistant in a household | Partner, son, and microcephalic child. She was pregnant at the moment of the interview*** | Christian | Low | None. But her husband had symptoms compatible with ZIKV when she was 5 months pregnant | All US exams were normal. In the last US exam, microcephaly was diagnosed. |
| **Consuelo** | 22 | Primary school | Child care | Partner, mother-in-law, father-in-law, and microcephalic child | Not declared | Low | Rash during pregnancy (month not specified) | First US exam was normal. During second US, health staff noticed baby's head was smaller than expected for gestational age. No diagnosis of microcephaly was done until baby was born. |

*(Continued)*

**Table 1.** (Continued)

| Name* | Age | Education | Occupation | Relatives living with her | Religion | SES **** | ZIKV symptoms (pregnancy month) | How was maternal ZIKV/ microcephalic children being diagnosed? |
|---|---|---|---|---|---|---|---|---|
| **Carmen** | 24 | Practical training | Nursing assistant** | Her microcephalic child | Christian | Low | Rash and fever (month not specified) | Microcephaly was detected by US. ZIKV screening in child blood samples were then performed. |

SES: Socio-economic status; US: Ultrasound.

*All names have been made up to guarantee anonymity.

**At the moment of the interview, she was on leave to take care of her child who had Dengue virus infection

***They live in a rural settlement as they were expelled from their home-town by the armed conflict in Colombia.

**** According to official government classification in Colombia, every district is rated from 1 to 6 for its affluence: very low (Status 1), low (Status 2), medium (Status 3), high (Status 4), very high (Status 5), and extremely high (Status 6). Populations living in districts of Status 1 and 2 are considered poor [23].

themselves from seeing images about babies with microcephaly during pregnancy; others expressed the importance of news on TV or the Internet to get informed about Zika, as quoting *"As far as you are, the information does not spread the same manner... what helps people more, is always TV"* (Judith, 29 years old). Some women expressed the importance of disseminating information about ZIKV to the general public to learn about the disease and how to prevent it, and to stakeholders to help affected families, as illustrated here:

*"I think that it* [information] *should not remain here like a research study, but it should be information that impacts, that arrives... arrives at other settings, mainly to the areas most at risk, for people to get to know this information. The fact that not knowing the information that is affecting us, indeed affects us, whether we are infected or not. But the fact of not knowing the information affects us. In general terms, first, because we are prone to... to live it, second, once we live it, we don't know how to handle it, thirdly, for all the children, let's say, it'll affect them because if you don't know how to handle it, you cannot take decisions, right? and the correct support"* (Judith, 29 years old).

Regarding preventive measures, many women felt resignation, and rage. The majority of them were already pregnant when recommendations of "postponing pregnancies" were done, leaving them hopeless: *"They started hanging postes in healthcare centers saying 'These are the symptoms. And then, 'Please, don't get pregnant!', but there were so many pregnant women already [...] Those who are pregnant cannot become "un-pregnant"* (Judith, 29 years old).

## Women's experiences and perceptions regarding their diagnosis of ZIKV infection and or the diagnosis of microcephaly in their babies

Most women declared that they were not ZIKV screened during pregnancy, even though they presented compatible symptoms. Those who were screened never received their results. In both cases, confirmation of maternal ZIKV infection was not possible. They expressed their concerns and feelings of *"rage"* about not having their results, to know the cause of their babies' congenital anomalies and severe manifestations such as microcephaly, seizures, arthrogryposis, eye and hearing problems. ZIKV diagnosis was based on clinical and epidemiological data. Women described common symptoms of ZIKV that they experienced during pregnancy (i.e. rash, fever, body pain). One participant suggested that women who had ZIKV infection in the third month of gestation (first trimester), had the most affected children, those suffering the worst consequences, including spastic paralysis, like her baby. One woman also explained

that ZIKV infection caused spontaneous abortions in her community as illustrated here: *"In most of the pregnancies. . . where I live. . . babies always died in the womb. They did not achieve to progress the pregnancy"* (Paola, 23 years old).

Each woman experienced a very different situation in regards to the diagnosis of anomalies in their children. Some women declared they were told about fetal anomalies during pregnancy *"one was informed about everything that was going to happen, and then one decided whether that was wanted or not"* (Paola, 23 years old). One participant declared that she was told that her baby had *"microcephaly"*, but she did not know what that term meant. She went home, told it to mother-in-law and she looked for the term on the Internet. Then, she felt fear when reading *"Children with microcephaly are those that have a small head, problems, and seizures"* (Jenny, 20 years old).

Other interviewees learned from it only once the baby was born *"Zika was a word I learned when [name of the child] was born"* (Judith, 29 years old). One participant explained that she was still on the operating table when the healthcare provider asked her if she had ZIKV as quoted: *"They did me a C-section. They took the baby when the doctor said 'Did you have Zika during pregnancy?' and my world turned upside down"* (María, 32 years old).

Some participants expressed the stigma and discomfort felt caused by some messages on how the idea of children born with CZS is drawn, as expressed in this quote: *"Basically, what was heard by society is that those children needed to be taken out from the womb [. . .] because they were going to be born like monsters! That was what they said. . . and sounded horrible [. . .] or that they would not live more than some months. . . that they were going to die"* (Adelaida, 27 years old).

## Discussions about a possible abortion

Regarding a possible termination of pregnancy, answers and opinions were very diverse; but all women would have appreciated more information about the course of their pregnancies', potential consequences of maternal ZIKV infection on their children, and to have been approached by healthcare professionals more skillfully to talk about such a sensitive topic. Some women were told that the baby had microcephaly while in the uterus, and could discuss a possible termination of pregnancy. Reasons for refusing it were based on their own decision, their families' decision, or religious beliefs. Sometimes, they felt *"almost forced to terminate the pregnancy"* or judged by healthcare staff and society if they decided to continue their pregnancy.

> *"What one hears from people [. . .] Why are you going to have a baby like this? Why?' They are so cruel. . . After I had her. . . 'Why did you have her? You're so selfish, you had the opportunity to avoid [your baby] having such a limited life'. . . and 'what makes you feel happy to have brought a baby who could never walk?' And lots of attacks. . ."* (Paola, 23 years old).

In other cases, women were not given the possibility to terminate their pregnancy. Some women hesitated about the fact that healthcare providers could have known in advance information about the health status of their fetuses, so they could have discussed different options. But most of the time, the diagnosis of anomalies came too late in pregnancy or after delivery. Misinformation and absence of options regarding SRHR were noted in participants interviews:

Interviewee: "At any time, did they tell you that there was a possibility to terminate the pregnancy?"

*Consuelo*: *"They told me it was too late [. . .] It was 6 months of pregnancy and that was not an option then. [. . .] We would indeed have interrupted the pregnancy because we were not*

*prepared for this. So, if we would have known it on time... we would have interrupted the pregnancy"* (Consuelo, 22 years old).

One woman explained that, even though she decided to continue her pregnancy, she would support other women and respect their personal decision of having, or not, having the baby. She said that this is a very difficult decision but needs to be addressed by each woman and her family, according to the circumstances in which they live, as these children would need special care throughout their lives. When prompted about what she would say about a woman whose fetus has been diagnosed with microcephaly, she stated *"Well, first... I would respect her decision, because this is something personal, because if she says she cannot have it, well, in my case, I would say I support her, I would say... I would comment on my situation with the child and so..."* (Carmen, 23 years old).

### Direct and indirect effects of carrying a child with microcephaly

During the interviews, women expressed the feelings they experienced while being diagnosed with ZIKV, and during childcare. Common feelings and emotions expressed were fear, rage, and sadness. Women talked about fear when they knew they were ZIKV infected, but also for the future, as most of them fear that they, or any other family member, could have another child with microcephaly. Rage was expressed mainly when revealing that they did not receive their ZIKV screening results, and the lack of support they received from local and national authorities. Sadness and suffering are mentioned when talking about their child's development, either because he/she is not able to walk, or if able, suffering not to hurt other mothers. One woman mentioned that her child is the only one capable of walking in this group of children, and when the other mothers see her child in the clinic, they start walking. This mum felt pity and worried about them, so she asked the doctor to attend to them first to avoid being in the waiting room with the other families with affected children. Happiness, and gratitude are also mentioned while talking about their children's development, religious beliefs, and hope to be heard and supported. Stigma was felt in certain ways throughout the interviews. Some families disappeared, as they did not want to have a relationship with the baby (sometimes, it was thought to be due to avoid future borrowing of money); even biological parents left one of these children with his grandmother. These aspects change behaviours. Some women expressed that during pregnancy they avoided watching TV because they received terrible images of babies with microcephaly. Once they were born, mothers had to change their daily routines to take care of their babies and monitor them all day and night, and they had to learn how to do specific care procedures at home, as almost all of them explained that they are not happy with the quality of the therapies provided by the health centre. These cross-cutting issues identified within all the themes discussed are detailed in Fig 1.

Most women, while talking about the difficulties faced raising a child with special needs, also mentioned how *"lovely and happy"* they and their children are; and that their existence is the main driver in their life. But also, a participant declared that they "were not ready" for assuming the challenges of caregiving a child with so many special needs. Some direct and indirect effects of caregiving a child with microcephaly in an endemic area for ZIKV, and a resource-constraint setting are detailed in Table 2.

### Main barriers and facilitators affecting adherence to medical follow-up of children with microcephaly

Mothers of children with microcephaly explained the huge range of defects their children present: neurodevelopmental delay, seizures, cerebral palsy, etc.; and, that they require specific drugs, vaccines, and therapies, to calm them, and eventually stand up and/or walk. Women

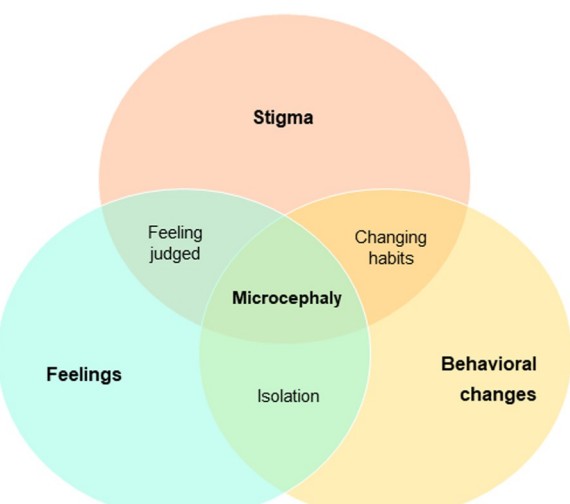

**Fig 1. Cross-cutting issues of first-hand experiences of mothers of microcephalic children.**

explained that they need to attend daily or weekly therapies in specific clinics, and try to do daily therapies at home. While the majority of women complained that those weekly visits are far from the ideal treatment their children needed, they also expressed the barriers they had to overcome to take them to the clinic.

One woman explained that microcephalic children's neurodevelopment highly depends on mothers' support, by all means (economic, educational, familiar. . .). To note, this family lives in the most socio-economically advanced household compared to the rest of the group; and they are aware that her child is the one who has developed the most (she can walk, attends school, and says a few words), quoted here:*"Children's improvement from all this process of therapy highly depends on mothers' support. For example, the fact that my husband supported me from day one is very important for me. He worried because if he had to look for a ticket for going to the city* [where clinical exams are being performed], *he would do it"* (Judith, 29 years old).

Additionally, one woman explained the great responsibility that is having to raise a baby with so many several anomalies, in a moment when still there are no specific treatments for children exposed to ZIKV while in the uterus, and their development is uncertain:*"We do not have the knowledge to carry out this huge responsibility, that sometimes, we don't know how to handle it"* (Adelaida, 27 years old).

The main barriers and facilitators detected affecting children's follow-up are described in Table 3.

After the interviews, all women appreciated the space for having had the possibility to talk about their processes of pregnancy, their children's health, and their family needs; as they did not perceive to have received psychosocial support or a time to talk to anybody about their experiences and feelings during these tough years:*"There is not any interest here [home country]. . . to say "let's help these children, let's see how mothers are feeling, let's support them. . . with a company, talk to them to see how they feel". Thank you for this. . . keep in mind these children, don't leave us alone because it's tough"* (Carla, undefined age).

## Discussion

The relevance of this study is vested in the rich in-depth information obtained on several topics raised by mothers of microcephalic children, who are still facing the consequences of the

**Table 2. Direct and indirect effects on the families with children with ZIKV-associated microcephaly according to participants.**

| Area | Sub-area | Highlights and quotes |
|---|---|---|
| **Individual spheres** | **Motivation** | • Change habits (wake up early and do exercise)<br>• Self-motivation to keep going (great spirit of overcoming)<br>• Start studying Nursing at University<br>*"My therapy is to get up early and exercise [. . .] because before, I was just crying, I didn't even wake up from bed [. . .] I said 'I'm going to wake up early every morning' and it's what has helped me the most. Because I go [running] from 4 to 5:30 am"* (María, 32 years old)<br>*"My family was not expecting that I had such a good score [for enrolling at University] because of the child. But I always had in mind that I had to fight for him, I had to have a good score for him [microcephalic child]. Everything that I have proposed to myself now is for him"* (Guadalupe, 18 years old)<br>*"Sometimes I feel alone because it's tough, every day the same routine; one gets bored. But I know that I decided to continue and the main support of [name of his baby] it's me!"* (Carla, undefined age) |
| | **Religiosity** | • Stronger religious beliefs<br>• Religion/spirituality was a coping mechanism for families to try to explain to themselves why they delivered a child with microcephaly<br>• Religion/spirituality was mentioned as an explanation to refuse a termination of pregnancy as it was believed to be *"God's will"*<br>• Religion/spirituality provided them strength to continue with their lives and take care of the baby<br>• Religion/spirituality makes them believe their children will improve (i.e. walk or talk)<br>*"Sometimes they say to you 'No, no, the girl is not going to walk'. The healthcare professional says one thing and God says another one* [because this child can walk]" (María, 32 years old)<br>*"For a reason God wanted to send him to us"* (Nadia, 21 years old)<br>*"God took the decision that she had to live"* (Judith, 29 years old) |
| **Social networks** | **Partner** | • More attached now, better relationship<br>• Does not want to take responsibilities on the baby<br>• Abandon them<br>• Started having unhealthy habits (alcohol abuse)<br>• Stopped them from having other children, until the microcephalic child is "self-sufficient"<br>• Stopped them from having other children because nobody assured them that microcephaly was a cause of Zika virus infection, and they are afraid of a congenital disease another child could inherit<br>*"I cannot complain, because I had support from my family, especially from his father [of the baby] because. . . [she starts crying] there are fathers that abandon their children. . . [crying] There are fathers that left mothers alone with their [microcephalic] babies, I saw it"* (Carla, undefined age) |
| | **Relatives** | • Lack of contact with relatives who do not accept the child<br>• Biological parents abandoned the child<br>• Support from their relatives<br>*"Sometimes there is not enough money, then they* [relatives] *always cooperate with us"* (María, 32 years old) |
| | **Other children within the family (siblings)** | • Worsening of mental health<br>• Does not feel prioritized<br>• Responsibility to take care of the baby when their mum is not at home or need to accompany them to therapies<br>• Made her change her mind in her future career, to study physical therapies to attend children<br>*"My older daughter is still really affected* [by having a microcephalic sister]. *I am going to look for a psychologist for her because sometimes she is normal, and others, she's crying, and crying, and crying for* [name of microcephalic child], *because she cannot walk, is not improving. . ."* (María, 32 years old)<br>*"It's tough for his sister to know that we give him* [microcephalic child] *lots of love, and not to her* [normocephalic sister]" (Adelaida, 27 years old) |
| | **Group of mothers with children with microcephaly** | • They meet with their children and see their progresses<br>• They talk and give advices to each other<br>• They practice sport together<br>*"Just by social media one talk to another, about the child that has something or another thing. . . but with time everything gets monotony, because they are all the same way, and so, the moment arrives that we don't talk about the babies, we don't know what to say [. . .] at least, we support each other, what for? 'We can do this, or this [to the baby]' among us. . . Because there's no support. . . governmental support, or from other people to help us, just us. If a child has something 'You can solve it with this' or a drug that he takes I say 'Well, this is happening, you can give this [to the baby]'"* (Adelaida, 27 years old) |

*(Continued)*

**Table 2.**  (Continued)

| Area | Sub-area | Highlights and quotes |
|---|---|---|
| **Socio-economic context** | **Economic** | Already poor families have decreased their incomes by several reasons:<br>• Not able to study or work outside the home and need to take special attention to the baby 24h a day. One interviewee sells refreshments and sweets at home, as that is the only thing she could do while staying home taking care of the child.<br>• Children need to attend several different medical specialists' different days and on different locations: paediatrics, infectiology, physiatry, among others<br>• Children need to attend neurodevelopmental therapies every day or at least twice a week<br>Hard decisions are taken by some families: 'to eat' one day or 'to go attend therapy'<br>• Some relatives refused to take care of the microcephalic child. One interviewee declared that her daughter and her son-in-law left their child with her (biological grandmother) and did not provide economic support either.<br>*"In therapy, they only give me 5000 pesos [to attend the visit]" [5000 Colombian pesos corresponds to 1.4 USD]"* (Nadia, 21 years old)<br>*"[The health centre] does not support us. They only support the children that need to come from outside the city"* (Carla, undefined age) |
| | **Psycho-social** | • Some woman received counselling, they talk to psychosocial workers while the children are in therapy<br>• Others, complain about the lack of psychosocial support available for these families<br>• Mental health of the mothers and other family members worsened during these years<br>*"During children's therapies there are psychologists that take some time for us also. They help us talking to express how we feel, we get relieved"* (Georgina, 43 years old)<br>*"One feels alone, sincerely, from the government, from society, and from a lot of people that surround us, sometimes even from family members"* (Adelaida, 27 years old) |

ZIKV epidemic. In Colombia, the National Healthcare Institute (INS) was in charge of analyzing all ZIKV samples, and updating epidemiological cases; but most results were not given back to patients. In Colombia, ZIKV definition relied on malformations found in fetal ultrasound and maternal clinical and/or epidemiological data. In our study, ZIKV maternal infection was not confirmed to be the cause of microcephaly due to lack of laboratory results. However, clinical and epidemiological data during pregnancy, and children clinical follow-up highly suggest it. All children in the study were born in 2016 when the ZIKV epidemic was at its peak, and laboratory resources and capacities were limited in endemic regions. Interviewees

**Table 3. Main barriers and facilitators affecting adherence to medical follow-up of microcephalic children.**

| Categories | Barriers | Facilitators |
|---|---|---|
| **Personal motivation** | • Not seeing that specialized care received is significantly improving children' neurodevelopment or abilities. | • Realising that the baby' neurodevelopment has improved with specialized care received. |
| **Social** | • Absence of partner/relatives' support.<br>• Difficulties in finding somebody to look for these babies, if mothers need to be outside from home, as they need specialized care. | • Relatives and neighbours sometimes care for children from other families.<br>• Close family/neighbours' relationships facilitate caregiving. |
| **Logistical** | • Some visits are very far away from their houses (even in different Districts): they lack private transportation, money to pay for public transport, or the baby is too heavy (3 years old) to be carried by their mothers' long kilometres.<br>• Most women carry their children for hours to get to the clinic. Weather conditions difficult attendance to therapies, especially in the rainy season. | • Referral to the nearest healthcare facility to increase attendance.<br>• Some women have been referred to centres close to their homes. In other cases, there are free buses from the clinic that pick up every mum and baby early morning to get to the therapies. |
| **Economic** | • Costs associated with specialized drugs and vaccines children need.<br>• Costs associated with weekly travel to therapies and monthly/yearly hospital-based exams (food for themselves and their accompaniment, loss of working hours/home chores/schooling for the mothers and their accompaniment...).<br>• Costs associated with impossibility to work outside the home with a dependent child.<br>• Lack of a baby trolley to facilitate child transport. | • Funds given for travel costs once they are in the clinic, they are more prone to attend those visits.<br>• Monetary compensation to buy essential needs such as diapers, but those require legal demands and a lot of bureaucracy. |

will never know if ZIKV was the cause of microcephaly, and this fact has an impact on their daily lives. These women carry the sorrow to remain forgotten by the healthcare system, public health agencies, and local, national, and international institutions. Interviewed women were very young, poor, and with different educational backgrounds. All of them were the biological mothers of children born with microcephaly in Caribbean Colombia in 2016, but one (grandmother but primary caregiver). Some women realized that the baby had microcephaly after delivery. Other women knew it, very late in pregnancy, and were offered a termination of pregnancy, but they refused it for different reasons. These children require special needs, which limits primary caregiver time and economic resources to study, work outside the home, take care of themselves, or other family members. Having a child with microcephaly changed women's trajectories, as most of them reported to be just caregivers due to difficulties in providing special needs and working outside the home.

Most women count on their partner's and relatives' support, and some found shelter in faith, sports, and their motivation to cope with difficulties. Women felt stigmatization, and a lack of support (rejection, incomprehension, judgment) in their external social environment, and sometimes even in the most intimate one, within their families. Stigma is related to those feelings experienced, as women felt judged on many occasions. A woman explained that even though she knew about fetal anomalies, she decided to continue her pregnancy, and her friends judged her and told her she was being "*selfish*" for bringing a child with such special needs. However, a feeling of sorority could be extracted from the interviews, as women talked to the others to get advice (i.e. to apply for specific therapies, monetary compensation, etc.), support each other, and see other children's progress. Local, national, and international support to cope with the negative economic, health, and psychosocial effects of the ZIKV pandemic in affected families is almost nonexistent.

Our results show different knowledge levels of ZIKV infection, disease, transmission, and prevention. Mostly, women with low levels of education were those with poor knowledge of the virus. Knowledge was also acquired by observations, as one participant correctly mentioned that ZIKV infections in the first trimester caused the most severe outcomes in children, after seeing other mothers in the study clinic. In line with other studies, women were not aware of the sexual transmission of the virus, thus, hampering the use of preventing measures [38–42]. Sexual and reproductive health education should address counselling on ZIKV, preventive methods, diagnosis, and management of the disease [42]. In our study, we noted a lack of SRHR campaigns that should have gone together with recommendations to avoid getting pregnant, along with assured access to contraceptive methods for women among all socio-economic strata. The low socio-economic status of women in this study could have influenced them for having a higher risk of ZIKV infection, and lower opportunities for being laboratory diagnosed on time, and to have had an ultrasonographic assessment to diagnose fetal microcephaly, as well as the possibilities to provide the special needs these children require. The fact that women declared that messages were prompted to women of reproductive age to avoid getting pregnant, highlighted that ZIKV was conceived as a problem of and for women. Some participants declared their incapacity to do anything as they were already pregnant. It was in line with mass media messages, that the responsibility was put on women to avoid ZIKV infection, without mentioning the role of men in the sexual transmission of the virus, and men's reproductive capacities and decisions [9,27,43,44]. Responses to public health crisis affecting SRHR, that ask women to postpone their pregnancies, are only feasible if decisions are made by women having complete information, and accessing comprehensive healthcare services within a human rights framework (modern contraception, avoid unplanned pregnancies, legal abortion services, etc.) [17]. According to a qualitative study performed with pregnant women in Puerto Rico, authors concluded that public health recommendations to avoid pregnancies

must consider how these messages were perceived within specific communities [45]. The prioritization of preventive measures that people could do, such as maintaining basic breeding hygiene and removing standing water, is essential for trust-building [45]. These findings are also applicable to our context in which trust in the information received was scarce.

Women in our study who were offered a termination of pregnancy refused to do it based on moral values, either personal or familiar ones and religious beliefs. On the contrary, also some women declared to have preferred to terminate the pregnancy in time if they would have known the anomalies their baby presented; but this was a hypothetical situation. Our results are in line with a recent study conducted in Colombia with 21 mothers of children with ZIKV-associated microcephaly, which found that abortion was not acceptable due to ethical concerns based on religiosity, the idea of maternity, and the patriarchal culture [46]. In our study, some healthcare providers imposed their social-conservative moral values around abortion saying to women that it was "*too late*" to terminate a pregnancy when the woman was still in the second trimester of pregnancy; and the legal regulations allow an interruption of pregnancy. There were several barriers to overcome to obtain a safe abortion in Colombia [26]. While legal at any gestational age, abortion remains highly stigmatized in the country, and only 11% of facilities eligible to offer it, actually do so [26]. Even though abortion is not criminalized in Colombia, social decriminalization has not happened yet. For abortions to become socially accepted, there is a great need for educational work, so that women could be informed about different options to choose from during pregnancy; and have the access to perform it. Efforts are needed to provide sufficient training to healthcare providers to sensitively approach women and not to impose healthcare staff moral values on women's reproductive health decisions. Educational work directed to women, men, and healthcare staff would have beneficial effects for people to freely decide about their reproductive decisions without feeling judged. Unfortunately, the ZIKV epidemic did not have an impact on the social acceptance of abortion, nor on policies to improve SRHR in countries where those were most needed. Unsafe abortions still affect the same women for whom access to sexual and reproductive health services was restricted [17]. Women were forced to face the challenges of giving birth to children with severe neurological malformations, increasing the already existing burden they faced before the epidemic, by their precarious and vulnerable living conditions [17].

According to a study that estimated the incidence of global unintended pregnancies, more than half of all pregnancies in Central and South America are unplanned, due to lack of access to good quality sexual and reproductive health education and services [41]. Differences in women's SRH access are related to intersectional inequalities; including increased risk of sexually transmitted infections, barriers to access quality primary healthcare, lack of adherence to protocols, stigma, and discrimination experienced by the poorest and most vulnerable women [16]. ZIKV amplified hierarchical structures and created inequalities. Stigmatization was not only present when women were infected by ZIKV, but they were judged because of their decisions, whether they continued, or not, their pregnancies, as seen in different interviews.

Besides, healthcare professionals need to provide coping strategies and social support for stress reduction, especially for those families living in low-resource settings [11]. Recent epidemics, such as ZIKV, have spread in poor countries, characterized by structural inequalities, high unemployment rates, poor sanitation, lower healthcare awareness, lack of healthcare access, mental health, etc., factors that contribute to viral spread and barriers that hinder adherence to treatment [47]. Commitments to improve mental health assessments and follow-up is especially important in women pregnant in a context of great uncertainty, and mothers of already affected children [17]. The plan to address the ZIKV epidemic should go beyond health-related interventions or vector control measures and should incorporate actions to address SRHR [16]. The recognition that the effects of the epidemic affect women and men

differently is necessary, along with recognition of ZIKV as a sexually transmitted disease, warrantee ZIKV screening diagnosis to pregnant women, access to contraception, and safe abortion [17]. An ethnographic study conducted in Colombia with mothers of microcephalic children highlighted the multiple negative changes that occur in the lives of their mothers, due to self-abandonment, mental and physical health outcomes [48].

All interviewees declared to have given birth by a C-section. This finding is in line with recent reports of increases in cesarean deliveries worldwide [49,50]. However, caesarean sections should not be routinely performed for the increased risks of maternal morbidity and mortality [50,51]. According to the WHO, these procedures should only represent 10–15% of all deliveries [50,51]. Studies performed in the USA, Italy, and Colombia have reported an increase in unjustified cesarean procedures in private clinics, a worrying concern due to the high costs and morbidity associated [22]. This malpractice is considered obstetric violence [52].

Children born with CZS have a broad range of long-term intellectual, physical, and sensory impairments [53], representing a wide-ranging impact on affected children, their families and society as a whole [54]. Microcephaly does not have a specific treatment because therapies vary depending on the degree of damage [19]. Ideal follow-up is done by a multidisciplinary team involving neurologists, neuropediatrician, speech therapy, physiotherapists, psychologist, occupational therapist, psychopedagoge, and other medical specialists [19]. Our study shows that families need to attend weekly visits to psychiatric therapies, and monthly/yearly visits to different medical services far from home and in different appointments, posing additional challenges for already poor families. Some barriers experienced by women are mainly related to lack of economic support to attend those visits, stipends for food, for somebody to accompany them, loss of productivity/work day, etc. Main facilitators for a correct follow-up of these children include monetary support for the visits, support from their couple, relatives, and the community.

All these barriers affect directly not only to mother and child but to the family system [55]. Duttine et al. conducted a meta-analysis about the needs of families of children with CZS and found that the caregivers of children with ZIKV-associated microcephaly experience challenges in mental health, health care access, and quality of life [55]. This is in line with the results found in the current study, within which the main barriers of families for achieving treatment adherence are healthcare access and quality of life. Duttine and colleagues found that financial hardships, difficulties with transport and services, and stigma, were other barriers faces by families [55]. Caregivers must pay transports to go to different specialists, to go to the hospital, to go to school -if children assist to school-; and, in some cases, caregivers must face stigma [55]. In the current study, one mother pointed out that she felt abandoned by the government, society, and even, by the family members. In the study of Romero-Acosta, et al, some mothers in Colombia also felt stigmatized by their own families [11]. This stigma could be perceived as abandonment from family members, above all, extended family.

Sometimes, religion acted as a coping mechanism for women to follow-up with their lives, but also healthy habits such as sport, and inner strength; as in line with other studies [11,38]. Support is also needed for concrete actions and material work for caring for these children [17]. Extensive evidence highlights that children with disabilities suffer from different exclusions related to poverty, malnutrition, vulnerability to violence, poor health, and school exclusion [54]. Difficulties will grow as adults with disabilities are less likely to be employed, and will face again poverty and social exclusion [54].

The main limitations of the study are that the sample size may not be big, although it does not represent a concern because of the richness of the information obtained. Regarding researchers' positionality, the interviewer was a Spanish white young woman, thus

participants' responses might have been influenced by the interviewer characteristics and status (nationality, employment, economic status, etc). Additionally, a feminist interpretation of Grounded Theory was used by the researcher to give special attention to issues related to oppression and gender inequality, such as daily caregiving of these children, impact in their lives, discussions about a possible termination of pregnancy, or other issues related to sexual and reproductive health. The way these positionings might have framed social desirability in participants' responses should not be disregarded. The main strengths of the study lie in the feedback and insight provided directly by affected populations; women who faced the challenges of ZIKV during the 2015–2017 epidemic. Qualitative results need to be interpreted with caution, as generalization cannot be performed. The benefits of speaking the same language (interviewees and interviewers) and being a female interviewer might have increased cooperation from participants who seemed comfortable, willing to share their experiences, and participate in-depth in the discussions. Further studies are needed to explore experiences in childcare, health access and decisions about sexual and reproductive health among other affected populations, including mothers (from diverse ethnic groups, rural-urban locations, migration and socio-economic statuses), fathers, siblings, other relatives of microcephalic children, and residents of the most affected communities.

## Conclusions

The ZIKV epidemic had devastating consequences on women of reproductive age. This study contributes to the understanding of the health inequities that ZIKV infection posed on women living in ZIKV affected areas, particularly among those living in areas of low socio-economic statuses, with structural inequalities, such as increased exposure to arboviruses, and lesser access to the healthcare system. Mothers of children born with microcephaly faced barriers that hindered them from accessing the specialized health care required for their children and the adequate psycho-social support for themselves. Most women had a high level of knowledge about ZIKV infection and preventive methods, though they were unaware of the sexual transmission of the virus. Their knowledge about ZIKV was acquired primarily through the mass media, the Internet, and the community, while the information provided by healthcare providers on ZIKV and its consequences was scarce or inexistent; including consideration of the possibility of pregnancy termination. It is of high importance to raise awareness of relevant stakeholders locally and globally to respond to all the public health problems including inequalities in access to the healthcare system, gender disparities, and violation of women's sexual and reproductive rights. Addressing social, medical, psychological, and economic needs faced by families with children heavily affected by ZIKV is essential to ensure access to the best possible health care, so that their children may achieve their greatest potential.

## Supporting information

**S1 Table. Summary of themes, and sub-themes emerging from the interviews.**
(DOCX)

## Acknowledgments

We are grateful to the women who participated in the study, their families, and children, for their patience and willingness to contribute to research studies, but mostly, their willingness to help other women.

## Author Contributions

**Conceptualization:** Elena Marbán-Castro, Maria Maixenchs, Azucena Bardají.

**Data curation:** Elena Marbán-Castro, Anna Marín-Cos.

**Formal analysis:** Elena Marbán-Castro, Cristina Enguita-Fernàndez, Anna Marín-Cos, Maria Maixenchs.

**Investigation:** Elena Marbán-Castro, Cristina Enguita-Fernàndez, Maria Maixenchs, Azucena Bardají.

**Methodology:** Elena Marbán-Castro, Cristina Enguita-Fernàndez, Kelly Carolina Romero-Acosta, Germán J. Arrieta, Anna Marín-Cos, Salim Mattar, Maria Maixenchs.

**Project administration:** Elena Marbán-Castro, Kelly Carolina Romero-Acosta, Germán J. Arrieta, Salim Mattar.

**Resources:** Cristina Enguita-Fernàndez, Kelly Carolina Romero-Acosta, Germán J. Arrieta, Salim Mattar, Maria Maixenchs, Azucena Bardají.

**Supervision:** Cristina Enguita-Fernàndez, Maria Maixenchs, Azucena Bardají.

**Validation:** Cristina Enguita-Fernàndez, Maria Maixenchs, Azucena Bardají.

**Writing – original draft:** Elena Marbán-Castro, Cristina Enguita-Fernàndez, Maria Maixenchs, Azucena Bardají.

**Writing – review & editing:** Elena Marbán-Castro, Cristina Enguita-Fernàndez, Kelly Carolina Romero-Acosta, Germán J. Arrieta, Anna Marín-Cos, Salim Mattar, Clara Menéndez, Maria Maixenchs, Azucena Bardají.

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
