## [Decision Letter · Decision Letter 0]

28 Jul 2021

Dear Marbán-Castro,

Thank you very much for submitting your manuscript "“One feels anger to know there is no one to help us!”: Perceptions of mothers of children with Zika-associated microcephaly in Caribbean Colombia: A qualitative study" for consideration at PLOS Neglected Tropical Diseases. As with all papers reviewed by the journal, your manuscript was reviewed by members of the editorial board and by several independent reviewers. In light of the reviews (below this email), we would like to invite the resubmission of a significantly-revised version that takes into account the reviewers' comments. 

We cannot make any decision about publication until we have seen the revised manuscript and your response to the reviewers' comments. Your revised manuscript is also likely to be sent to reviewers for further evaluation.

Sincerely,

Ran Wang, Ph.D., M.D.

Associate Editor

Dennis Bente

Deputy Editor

Reviewer's Responses to Questions

**Key Review Criteria Required for Acceptance?**

**Methods**

-Are the objectives of the study clearly articulated with a clear testable hypothesis stated?

-Is the study design appropriate to address the stated objectives?

-Is the population clearly described and appropriate for the hypothesis being tested?

-Is the sample size sufficient to ensure adequate power to address the hypothesis being tested?

-Were correct statistical analysis used to support conclusions?

-Are there concerns about ethical or regulatory requirements being met?

Reviewer #1: The aims and methods were clearly described. I only have one minor suggestion in the "Abstract" to include the number of mothers interviewed in the methodology description. (it was on the author summary);

Reviewer #2: Dear Authors, 

I started reading with great interest your article on a topical and important problem that arose after the end of the Zika pandemic, but later I found out that some important corrections need to be made to highlight the value of the article and I present them in detail below.

INTRODUCTION 

• In this section is important to describe in more detail the CZS, the comorbid disorders, as well as its serious effects on the neurodevelopment of infants and children. 

• Furthermore, if you have access, it would be great to mention the average monthly income in Colombia, the monthly expenses of a family with a child with CZS and how much the state compensates these families could also be included in a short paragraph. 

• Also, it should be mentioned the percentage of people who are uninsured or living below the poverty line.

• The insurance system in Colombia

METHODS 

• In the CZS, the spectrum of disability varies depending on the severity of microcephaly. Why was your sample not grouped according to their degree of disability?

Reviewer #4: These were described adequately - although there was little description of how participants were selected and recruited. There is also limited discussion of the positionality of the researchers.

**Results**

-Does the analysis presented match the analysis plan?

-Are the results clearly and completely presented?

-Are the figures (Tables, Images) of sufficient quality for clarity?

Reviewer #1: The manuscript is well-written and deals with a very important topic focused on perceptions of mothers of children with Congenital Zika Syndrome. 

The results point to a large spectrum based on knowledge about health care, assistance and rights; and some crucial social, subjective, and economic aspects that were affected in their lives. 

I have minor comments:

- I liked how you used the tables and figure to describe and summarize the results;

- Table 2 - include the word "direct" in the description of the table. (Direct and indirect effects...);

Follow few comments and questions, in case you think they are relevant to your argument in results and discussion:

- The fact of participants live in rural and peri-urban areas with low socio-economic status could be an influence of these mothers' views/perceptions? Could you explore it more in the discussion? Are there studies like this in urban areas in Colombia? 

- Some studies in Brazil point that groups of mothers with children with microcephaly whether it is online (like WhatsApp groups) or officially organized associations are an interesting way to struggle by rigts, share lived experiences, and have a kind of social support. Have you seen it in your data? I would like to know more about how the mothers share with each other. 

- Most of the mothers interviewed had as occupation "caring of her child". Was it a given need by had a child with CZS? To have a child with a CZS has changed their trajectories and project of lives?

Reviewer #2: The results presented clearly and anatically 

The tables have sufficient quality 

Please explain ''What do you define as “low Socio-economic status?''

Reviewer #4: There are many linguistic errors, some of which I have highlighted. The manuscript should be careful read through and edited to address these.

The results are presented as themes and described in text, tables and a diagram. It is adequate

**Conclusions**

-Are the conclusions supported by the data presented?

-Are the limitations of analysis clearly described?

-Do the authors discuss how these data can be helpful to advance our understanding of the topic under study?

-Is public health relevance addressed?

Reviewer #1: (No Response)

Reviewer #2: DISCUSION 

• You refer the cesarean section as a routine procedure in cases with microcephaly. Can you tell if you have access, the rates of cesarean sections in Colombia?

• The syndrome has a significant psychosocial impact on families, especially mothers and consequently health care providers. Can you report similar studies from all the affected areas?

CONCLUSIONS 

• At this point it is important to emphasize the need to raise awareness of relevant stakeholders, not only locally but also globally. This article highlights not only a significant public health problem and the inequality in access to health services, it presents significant gender inequality issues through presents significant problems of gender inequality through the violation of women's rights and their self-disposition.

Reviewer #4: Overall the conclusions were supported well by the data presented and the limitations given. There was perhaps more that could have been done to explore the positionality of the researchers and how this might have influenced the interviews and the analysis of the data. 

A conclusion that the Zika virus has differential affect on men and women is hard to support without having included men in the research.

**Editorial and Data Presentation Modifications?**

Reviewer #1: (No Response)

Reviewer #2: Dear Editor 

Thank you for the invitatation to review this manuscript. I suggest minor revisions 

Kind regards

Reviewer #4: The text includes a great many grammatical and linguistic errors. The text would need to be carefully edited to address these.

**Summary and General Comments**

Reviewer #1: (No Response)

Reviewer #2: INTRODUCTION 

• In this section is important to describe in more detail the CZS, the comorbid disorders, as well as its serious effects on the neurodevelopment of infants and children. 

• Furthermore, if you have access, it would be great to mention the average monthly income in Colombia, the monthly expenses of a family with a child with CZS and how much the state compensates these families could also be included in a short paragraph. 

• Also, it should be mentioned the percentage of people who are uninsured or living below the poverty line.

• The insurance system in Colombia

METHODS 

• In the CZS, the spectrum of disability varies depending on the severity of microcephaly. Why was your sample not grouped according to their degree of disability?

RESULTS 

• What do you define as “low Socio-economic status”? 

DISCUSION 

• You refer the cesarean section as a routine procedure in cases with microcephaly. Can you tell if you have access, the rates of cesarean sections in Colombia?

• The syndrome has a significant psychosocial impact on families, especially mothers and consequently health care providers. Can you report similar studies from all the affected areas?

CONCLUSIONS 

• At this point it is important to emphasize the need to raise awareness of relevant stakeholders, not only locally but also globally. This article highlights not only a significant public health problem and the inequality in access to health services, it presents significant gender inequality issues through presents significant problems of gender inequality through the violation of women's rights and their self-disposition.

Reviewer #4: This is an important study exploring the views and experiences of mothers or primary caregivers caring for an infant that has Sika associated microcephaly in Caribbean Colombia. The weakness of the paper that need to be addressed are: improved description of their methods including recruitment, translation in transcription or at report writing and the positionality of the researchers. There are numerous errors, some of which I have highlighted on the pdf of the manuscript which need to be corrected.
---

## [Decision Letter · Decision Letter 1]

29 Nov 2021

Dear Marbán-Castro,

Thank you very much for submitting your manuscript "“One feels anger to know there is no one to help us!”: perceptions of mothers of children with Zika-associated microcephaly in Caribbean Colombia: A qualitative study" for consideration at PLOS Neglected Tropical Diseases. As with all papers reviewed by the journal, your manuscript was reviewed by members of the editorial board and by several independent reviewers. In light of the reviews (below this email), we would like to invite the resubmission of a significantly-revised version that takes into account the reviewers' comments. 

We cannot make any decision about publication until we have seen the revised manuscript and your response to the reviewers' comments. Your revised manuscript is also likely to be sent to reviewers for further evaluation.

Sincerely,

Ran Wang, Ph.D., M.D.

Associate Editor

Dennis Bente

Deputy Editor

Reviewer's Responses to Questions

**Key Review Criteria Required for Acceptance?**

**Methods**

-Are the objectives of the study clearly articulated with a clear testable hypothesis stated?

-Is the study design appropriate to address the stated objectives?

-Is the population clearly described and appropriate for the hypothesis being tested?

-Is the sample size sufficient to ensure adequate power to address the hypothesis being tested?

-Were correct statistical analysis used to support conclusions?

-Are there concerns about ethical or regulatory requirements being met?

Reviewer #1: The aims and methods were clearly described and more details about participants' recruitment were pointed.

Reviewer #2: • In the CZS, the spectrum of disability varies depending on the severity of microcephaly. Why was your sample not grouped according to their degree of disability?

Reviewer #4: This study has used qualitative methods to explore the experiences of women who have had a child affected by the Zika virus. The design is appropriate and the authors have highlighted important findings. The study authors did not seek to get a sample that represented different socioeconomic groups or ensure that their sample was representative in terms of certain factors. This would have strengthened the work. 

There are some important findings, particularly the way the Zika infections exacerbated inequalities, the poor access to family planning services, the lack of access to information, the stigma experienced by women. The findings give valuable information for ways that future outbreaks can be managed and what information needs to be disseminated to vulnerable populations and the nature of the support needed for women in these communities.

**Results**

-Does the analysis presented match the analysis plan?

-Are the results clearly and completely presented?

-Are the figures (Tables, Images) of sufficient quality for clarity?

Reviewer #1: The manuscript is well-written and deals with a very important topic focused on perceptions of mothers of children with Congenital Zika Syndrome.

The results point to a large spectrum based on knowledge about health care, assistance, gender and rights; and some crucial social, subjective, and economic aspects that were affected in their lives.

The tables and figure describe and summarize the results in a good way.

Reviewer #2: The results presented clearly and anatically

The tables have sufficient quality

Please explain ''What do you define as “low Socio-economic status?''

Reviewer #4: The results are clearly presented. the paper has a number of grammatical errors and in two sections the analysis and conclusions do not feel that they are well embedded in the findings. I have made these clear in my accompanying text.

**Conclusions**

-Are the conclusions supported by the data presented?

-Are the limitations of analysis clearly described?

-Do the authors discuss how these data can be helpful to advance our understanding of the topic under study?

-Is public health relevance addressed?

Reviewer #1: (No Response)

Reviewer #2: CONCLUSIONS

• At this point it is important to emphasize the need to raise awareness of relevant stakeholders, not only locally but also globally. This article highlights not only a significant public health problem and the inequality in access to health services, it presents significant gender inequality issues through presents significant problems of gender inequality through the violation of women's rights and their self-disposition.

Reviewer #4: There are two conclusions where I feel the authors have drawn conclusions that are not supported by the data. The final sentence of the abstract is also vague and could be far more informative given the richness of the findings.

**Editorial and Data Presentation Modifications?**

Reviewer #1: (No Response)

Reviewer #2: Dear Editor, 

Unfortunately, the authors did not revise the manuscript according to our suggestions.

Reviewer #4: These are detailed on an attached word document

**Summary and General Comments**

Reviewer #1: (No Response)

Reviewer #2: INTRODUCTION

• In this section is important to describe in more detail the CZS, the comorbid disorders, as well as its serious effects on the neurodevelopment of infants and children.

• Furthermore, if you have access, it would be great to mention the average monthly income in Colombia, the monthly expenses of a family with a child with CZS and how much the state compensates these families could also be included in a short paragraph.

• Also, it should be mentioned the percentage of people who are uninsured or living below the poverty line.

• The insurance system in Colombia

METHODS

• In the CZS, the spectrum of disability varies depending on the severity of microcephaly. Why was your sample not grouped according to their degree of disability?

The results presented clearly and anatically

The tables have sufficient quality

Please explain ''What do you define as “low Socio-economic status?''

DISCUSION

• You refer the cesarean section as a routine procedure in cases with microcephaly. Can you tell if you have access, the rates of cesarean sections in Colombia?

• The syndrome has a significant psychosocial impact on families, especially mothers and consequently health care providers. Can you report similar studies from all the affected areas?

CONCLUSIONS

• At this point it is important to emphasize the need to raise awareness of relevant stakeholders, not only locally but also globally. This article highlights not only a significant public health problem and the inequality in access to health services, it presents significant gender inequality issues through presents significant problems of gender inequality through the violation of women's rights and their self-disposition.

Dear Editor

Thank you for the invitatation to review this manuscript. I suggest minor revisions

Kind regards

INTRODUCTION

• In this section is important to describe in more detail the CZS, the comorbid disorders, as well as its serious effects on the neurodevelopment of infants and children.

• Furthermore, if you have access, it would be great to mention the average monthly income in Colombia, the monthly expenses of a family with a child with CZS and how much the state compensates these families could also be included in a short paragraph.

• Also, it should be mentioned the percentage of people who are uninsured or living below the poverty line.

• The insurance system in Colombia

METHODS

• In the CZS, the spectrum of disability varies depending on the severity of microcephaly. Why was your sample not grouped according to their degree of disability?

RESULTS

• What do you define as “low Socio-economic status”?

DISCUSION

• You refer the cesarean section as a routine procedure in cases with microcephaly. Can you tell if you have access, the rates of cesarean sections in Colombia?

• The syndrome has a significant psychosocial impact on families, especially mothers and consequently health care providers. Can you report similar studies from all the affected areas?

CONCLUSIONS

• At this point it is important to emphasize the need to raise awareness of relevant stakeholders, not only locally but also globally. This article highlights not only a significant public health problem and the inequality in access to health services, it presents significant gender inequality issues through presents significant problems of gender inequality through the violation of women's rights and their self-disposition.

Reviewer #4: There are some important findings, particularly the way the Zika infections exacerbated inequalities, the poor access to family planning services, the lack of access to information, the stigma experienced by women. The findings give valuable information for ways that future outbreaks can be managed and what information needs to be disseminated to vulnerable populations and the nature of the support needed for women in these communities.

The weakness in the paper include some improvements in grammatical errors made

The abstract needs strengthening

The interpretation of the action of health care professionals in relation to abortion, and the comments about the high c section rate stood out as observations that were not substantiated by the data and rather reflected the positionality of the authors.

These are major revisions that need to be addressed before this important paper can be published.

PLOS authors have the option to publish the peer review history of their article (what does this mean?). If published, this will include your full peer review and any attached files.

Reviewer #1: No

Reviewer #2: No

Reviewer #4: No
---

## [Decision Letter · Decision Letter 2]

14 Mar 2022

Dear Marbán-Castro,

We are pleased to inform you that your manuscript '“One feels anger to know there is no one to help us!”. Perceptions of mothers of children with Zika virus-associated microcephaly in Caribbean Colombia: A qualitative study' has been provisionally accepted for publication in PLOS Neglected Tropical Diseases.

Best regards,

Ran Wang, Ph.D., M.D.

Associate Editor

Dennis Bente

Deputy Editor

Reviewer's Responses to Questions

**Key Review Criteria Required for Acceptance?**

**Methods**

-Are the objectives of the study clearly articulated with a clear testable hypothesis stated?

-Is the study design appropriate to address the stated objectives?

-Is the population clearly described and appropriate for the hypothesis being tested?

-Is the sample size sufficient to ensure adequate power to address the hypothesis being tested?

-Were correct statistical analysis used to support conclusions?

-Are there concerns about ethical or regulatory requirements being met?

Reviewer #2: The methods presendet clearly and analitically

Reviewer #5: (No Response)

**Results**

-Does the analysis presented match the analysis plan?

-Are the results clearly and completely presented?

-Are the figures (Tables, Images) of sufficient quality for clarity?

Reviewer #2: The results presendet clearly

Reviewer #5: (No Response)

**Conclusions**

-Are the conclusions supported by the data presented?

-Are the limitations of analysis clearly described?

-Do the authors discuss how these data can be helpful to advance our understanding of the topic under study?

-Is public health relevance addressed?

Reviewer #2: the conclusions are clearly

Reviewer #5: (No Response)

**Editorial and Data Presentation Modifications?**

Reviewer #2: (No Response)

Reviewer #5: (No Response)

**Summary and General Comments**

Reviewer #2: Most issues have been overtaken

Reviewer #5: The epidemic of Zika virus has drawn global attention, largely due to its association with the dramatic increase of neonatal microcephaly. In this study, Elena Marbán-Castro et al. used qualitative methods to investigate the perceptions about ZIKV infection among mothers of children born with microcephaly during the ZIKV epidemic in Caribbean Colombia, and the barriers and facilitators affecting child health follow-up. The design is appropriate and the findings are important. These findings will give valuable information for ways that future outbreaks can be better managed and what information and supports need to be provided to vulnerable populations in these communities.

The authors have satisfactorily answered all the comments raised by previous reviewers, and therefore the manuscript may be accepted for publication with only very minor modifications as listed below.

1. Page 13, Line 283 “Other interviewees learned from itonly once the baby was born...”, should be “...it only...”

PLOS authors have the option to publish the peer review history of their article (what does this mean?). If published, this will include your full peer review and any attached files.

---

## [Editor Report · Acceptance letter]

1 Apr 2022

Dear Marbán-Castro,

We are delighted to inform you that your manuscript, "“One feels anger to know there is no one to help us!”. Perceptions of mothers of children with Zika virus-associated microcephaly in Caribbean Colombia: A qualitative study," has been formally accepted for publication in PLOS Neglected Tropical Diseases.

Best regards,

Shaden Kamhawi

co-Editor-in-Chief

Paul Brindley

co-Editor-in-Chief
